# Temporal Distribution Patterns of Cryptic *Brachionus calyciflorus* (Rotifera) Species in Relation to Biogeographical Gradient Associated with Latitude

**DOI:** 10.3390/ani14020244

**Published:** 2024-01-12

**Authors:** Yuan Xu, Le-Le Ge, Xin-Feng Cheng, Xian-Ling Xiang, Xin-Li Wen, Yong-Jin Wang, Hao Fu, Ya-Li Ge, Yi-Long Xi

**Affiliations:** 1School of Ecology and Environment, Anhui Normal University, Wuhu 241002, China; 13955604898@163.com (Y.X.); gelele1996@163.com (L.-L.G.); chxf0553@ahnu.edu.cn (X.-F.C.); xiangxianling@163.com (X.-L.X.); wenxinli1977@126.com (X.-L.W.); 2Collaborative Innovation Center of Recovery and Reconstruction of Degraded Ecosystem in Wanjiang Basin Co-Funded by Anhui Province and Ministry of Education of the People’s Republic of China, Anhui Normal University, Wuhu 241002, China; 3Management Committee of Scenic Attraction of Lake Yunlong, Xuzhou 221007, China; xzslrsc@126.com; 4Reservoir Management Office of Lake Yunlong, Xuzhou 221007, China; fuhao19810203@163.com

**Keywords:** rotifers, cryptic species, seasonal succession, temporal overlap, environmental variables, climatic zone

## Abstract

**Simple Summary:**

In subtropical shallow lakes, large-scale changes in water temperature lead to seasonal succession of some cryptic rotifer species, and the temporal overlap of other cryptic rotifer species is a common phenomenon. However, in tropical shallow lakes, relatively stable water temperatures throughout the year may not lead to a seasonal succession of cryptic rotifer species, but evidence is scarce. Studies on the temporal distribution patterns of the cryptic *Brachionus calyciflorus* species in three lakes in China revealed that in the warm-temperate Lake Yunlong, *B. fernandoi* and *B. calyciflorus* s.s. underwent a seasonal succession, which was largely attributed to their differential adaptation to water temperature. In the subtropical Lake Jinghu, *B. fernandoi*, *B. calyciflorus* s.s., and *B. dorcas* exhibited both seasonal succession and temporal overlap. Seasonal successions were largely attributed to their differential adaptation to temperature, and temporal overlap resulted from their differential responses to algal food concentration. In the tropical Lake Jinniu, *B. calyciflorus* s.s. persisted throughout the year and overlapped with *B. dorcas* for 5 months. The temporal overlap resulted from their differential responses to copepod predation. These results indicated that the temporal distribution pattern of the cryptic *B. calyciforus* species and the mechanism that allows competitor coexistence vary with different climate zones.

**Abstract:**

Sympatric distribution and temporal overlap of cryptic zooplankton species pose a challenge to the framework of the niche differentiation theory and the mechanisms allowing competitor coexistence. We applied the methods of phylogenetic analysis, DNA taxonomy, and statistical analysis to study the temporal distribution patterns of the cryptic *B. calyciflorus* species, an excellent model, in three lakes, and to explore the putative mechanisms for their seasonal succession and temporal overlap. The results showed that in the warm-temperate Lake Yunlong, *B*. *fernandoi* and *B*. *calyciflorus* s.s. underwent a seasonal succession, which was largely attributed to their differential adaptation to water temperature. In the subtropical Lake Jinghu, *B*. *fernandoi*, *B*. *calyciflorus* s.s., and *B. dorcas* exhibited both seasonal succession and temporal overlap. Seasonal successions were largely attributed to their differential adaptation to temperature, and temporal overlap resulted from their differential responses to algal food concentration. In the tropical Lake Jinniu, *B*. *calyciflorus* s.s. persisted throughout the year and overlapped with *B. dorcas* for 5 months. The temporal overlap resulted from their differential responses to copepod predation. These results indicated that the temporal distribution pattern of the cryptic *B. calyciforus* species and the mechanism that allows competitor coexistence vary with different climate zones.

## 1. Introduction

In past two decades, the use of DNA taxonomy and integrative approaches combining morphological, ecological, and molecular data have revealed the existence of cryptic species in a broad range of taxonomic groups [1,2,3,4,5]. As species belonging to the same cryptic species complex are so similar in their morphology and physiology, a high degree of ecological similarity and, hence, competitive exclusion is expected to occur between them [6,7]. However, cryptic species commonly exist in sympatry [6,8,9,10], which poses a challenge to the framework of the niche differentiation theory and the mechanisms that allow competitor coexistence [7,11,12,13].

Phylum Rotifera is one of the groups of animals with the highest level of occurrence of cryptic species complexes. Up to now, 54 cryptic rotifer species complexes have been discovered [13]. Among them, the euryhaline *Brachionus plicatilis* species complex has been the subject of many studies on the temporal distribution patterns of cryptic rotifer species and the mechanisms that allow competitor coexistence. Previous studies have shown that the temporal distribution of the cryptic *B. plicatilis* species generally displays both seasonal succession and temporal overlap [8,14,15,16,17,18]. Seasonal succession is largely attributed to their differential adaptation to salinity and/or temperature [8,14,15,16,17,18,19,20,21], and temporal overlap results from their differential responses to environmental conditions such as salinity [8,14,15,16,20,21] and oxygen availability [22], resource partitioning, and differential vulnerability to predators [23,24,25].

The freshwater *B. calyciflorus* species complex—which was recently suggested to consist of four species, *B. dorcas*, *B. elevatus*, *B. calyciflorus* s.s., and *B. fernandoi* [26]—has also received attention in studies on the temporal distribution patterns of cryptic rotifer species and the mechanisms allowing competitor coexistence. The studies on this species complex inhabiting a warm-temperate pond and several subtropical shallow lakes have revealed that the temporal distribution of the cryptic *B. calyciflorus* species generally displays both seasonal succession and temporal overlap [27,28,29,30,31,32,33]. The warm-temperate ponds and subtropical shallow lakes inhabited by the *B. calyciflorus* complex have low spatial heterogeneity but high temporal variability, of which the most obvious is the seasonal variation of water temperature [27,28,29,30,31]. In tropical shallow lakes, however, relatively stable water temperatures throughout the year may not lead to seasonal succession of the cryptic *B. calyciflorus* species, but evidence is scarce. To further explore the role of temperature in shaping the occurrence and distribution of the species within the *B. calyciforus* complex, it would be worthwhile to investigate the possibility of a biogeographical gradient associated with latitude, which could be connected to variations in water temperature [13].

In this study, we applied the methods of phylogenetic analysis, DNA taxonomy, and principal component analysis to investigate the temporal distribution patterns of the cryptic *B. calyciflorus* species in three lakes in China—the warm-temperate Lake Yunlong, subtropical Lake Jinghu, and tropical Lake Jinniu—and explore the putative mechanisms for their seasonal succession and/or temporal overlap. We tested the following hypotheses: (i) the temporal distribution pattern of the cryptic *B. calyciforus* species varies with different climate zones; (ii) the mechanisms underlying the temporal overlap of potentially strong competitors are different between climate zones.

## 2. Materials and Methods

### 2.1. Sample Collection and Environment Variables Analyses

Zooplankton samplings were carried out monthly in Yunlong, Jinghu, and Jinniu lakes from October 2018 to September 2019. Lake Yunlong (34.24° N, 117.17° E) is located in Xuzhou City, Jiangsu Province, and has a surface area of 6.76 km^2^ and an average water depth of 2.5 m. Lake Jinghu (31.33° N, 118.37° E) is located in Wuhu City, Anhui Province, and has a surface area of 0.15 km^2^ and an average water depth of 1.5 m. Lake Jinniu (20.01° N, 110.32° E) is located in Haikou City, Hainan Province, and has a surface area of 1.98 km^2^ and an average water depth of 2.0 m. On each occasion, two quantitative zooplankton samples were obtained from two fixed sites of each lake by filtering (25 μm mesh net) two samples of integrated lake water (5 L of water from the surface to the bottom at 0.5 m intervals) and fixing in situ with 4% formaldehyde. From these quantitative samples, the density estimates of the *B. calyciflorus* species complex and its potential competitors and predators, including cladocerans, omnivorous rotifers (e.g., *Asplanchna* spp.), and copepods, were calculated using direct counts of females under an Olympus BH-2 microscope with 100× magnification. Additional two qualitative zooplankton samples were also collected at the fixed sites of each lake in several hauls using a 25-µm plankton net, fixing in situ with 90% ethanol, and then transported to the laboratory. Under a stereomicroscope, individuals belonging to the *B. calyciflorus* species complex were isolated from each sample, washed several times with double distilled water, and preserved at −20 °C until molecular processing.

Simultaneously with the collection of zooplankton samples, water temperature, pH value, and dissolved oxygen (DO), total nitrogen (TN), total phosphorus (TP), and ammonium–nitrogen (NH_4_^+^–N) concentrations were measured, as described in details by Wen et al. [34]. Integrated lake water was collected using a 5-l modified Van-Dorn sampler (5 L of water from the surface to the bottom at 0.5 m intervals) and then filtered through a 25-μm mesh net. Filtrate was further filtered through Whatman GF/C glass-fiber filters (0.45 µm pore size) to obtain relatively small phytoplankton that might be a food resource for rotifers. The biomass of relatively small phytoplankton was represented by chlorophyll *a* (Chl-*a*) concentration. Chl-*a* content was spectrophotometrically measured after extracting the filters overnight in darkness using 90% acetone and calculated without correcting for phaeopigments [35].

### 2.2. DNA Extraction, PCR Amplification and Sequencing

The HotSHOT technique [36] was used to extract DNA from individual rotifers. An individual rotifer was transferred into a 0.2 mL EP tube containing 30 μL of alkaline lysis buffer under a stereomicroscope. Once in the buffer, the rotifer was crushed against the side of the tube using a sterile pipette tip. The sample was incubated at 95 °C for 30 min and stored on ice for 3–4 min. After a further 30 µL of neutralizing buffer was added to the EP tube, the sample was vortexed briefly and spun down, and then stored at −20 °C.

PCR amplification was conducted using the iCycler PCR machine (Bio-Rad Research Company, Hercules, CA, USA). The primers for the mitochondrial cytochrome c oxidase subunit I (mtCOI) gene sequence were HCO2198 (5′-TAAACTTCAGGGTGACCAAAAAATCA-3′) and LCO1490 (5′-GGTCAACAAATCA TAAAGATATTGG-3′) [37], and those for the nuclear internal transcribed spacer 1 locus (nuITS1) sequence were LH2 (5′-GTCGAATTCGTAGGTGAACCTGCGGAAGGATCA-3′) and Dlam (5′-CCTGCAGTCGACAKATGCTTAARTTCAGCRGG-3′) [38]. All reagents and primers were obtained from Sangon Biotechnology Co. Ltd. (Shanghai, China). A 25-μL amplification system for mtCOI sequences consisted of 2.5 μL of 10 × PCR buffer, 0.5 μL of each primer (0.01 mM), 2 μL of MgCl_2_ (25 mM), 2 μL of each dNTP (25 mM), 5.0 μL of template DNA, and 0.4 μL of *Taq* DNA polymerase (Takara, Tokyo, Japan). Amplification of the mtCOI sequence was performed using the following cycling conditions: pre-denaturation at 95 °C for 5 min and 35 cycles of denaturation at 94 °C for 40 s, annealing at 48 °C for 30 s, elongation at 72 °C for 2 min, and final extension at 72 °C for 20 min. A 25-μL amplification system for nuITS1 sequences consisted of 2.5 μL of 10 × PCR buffer, 0.5 μL of each primer (0.01 mM), 2 μL of MgCl_2_ (25 mM), 2 μL of each dNTP (25 mM), 5.0 μL of template DNA, and 0.5 μL of *Taq* DNA polymerase (Takara). Amplification of the nuITS1 sequence was performed using the following cycling conditions: pre-denaturation at 95 °C for 5 min and 35 cycles of denaturation at 94 °C for 30 s, annealing at 48 °C for 30 s, elongation at 72 °C for 1 min and final extension at 72 °C for 10 min. After electrophoresis on 0.8% agarose gels, the PCR products were sequenced using an ABI-PRISM 3730 automated sequencer.

### 2.3. Sequence Alignment and Phylogenetic Analyses

All sequences were aligned individually using the default settings of the online version (http://blast.ncbi.nlm.nih.gov/Blast.cgi, accessed on 5 November 2021) of BLAST [39]. Fragments of 632 bp and 270 bp were selected as the mtCOI and nuITS1 target sequences, respectively.

The phylogenetic relationships were reconstructed using two optimality criteria: Maximum Likelihood (ML) and Bayesian Inference (BI). The most optimal sequence evolution parameters and models (TVMþG and GTRþG), as selected by Modeltest 3.7 [40], were used as settings in PAUP and Bayesian phylogenetic analyses based on the mtCOI and nuITS1 sequences. Two independent Bayesian analyses with the Markov Chain Monte Carlo (MCMC) method were conducted in MrBayes 3.1.2 [41], with four chains per analysis and randomly chosen starting trees. The Markov chains were run for 10,000,000 generations, with trees sampled every 100 generations. The first 250,000 generations were discarded as burn-in, and the remaining trees were used to estimate Bayesian posterior probabilities. In order to discriminate the lineage relationships between COI/ITS1 groups within the *B. calyciflorus* species complex found in this study, which was demonstrated by Xiang et al. [42,43] and Papakostas et al. [44], five mtCOI and nuITS1 sequences of *B. calyciflorus* species complex were obtained from GenBank (the accession number of the five mtCOI sequences are AQ_W13_GU232548, DZ_W2_GU232575, TJ_S10_FJ826940, WH_S23_FJ826934 and XZ_W2_GU232725; accordingly, those five nuITS1 sequences are AQ_W13_FJ937455, DZ_W2_FJ937482, TJ_S10_GU012757, WH_S23_GU012785 and XZ_W2_FJ937632) and used for phylogenetic analysis. The sequences of a cryptic *B. plicatilis* species (GenBank accession number of the mtCOI and nuITS1 sequences are JX293046 and KU299746) were used as outgroup in the phylogenetic reconstruction based on the mtCOI and nuITS1 sequences, respectively.

### 2.4. COI/ITS1 Group Diagnosis and Abundance Estimation

Three main types of species-delimitation methods, including the Automatic Barcode Gap Discovery (ABGD), Poisson Tree Process (PTP), and Generalized Mixed Yule Coalescent (GMYC) models [45,46], were applied to explore the number of reproductively isolated COI/ITS1 groups in the *B. calyciflorus* species complex. The ABGD model (available from http://wwwabi.snv.jussieu.fr/public/abgd/abgdweb.html, accessed on 12 April 2009) was applied to automatically discover the barcode gap instead of using one or several predefined distance thresholds for COI/ITS1 group delimitation [47]. The PTP model was applied to the input ML trees using coalescence theory to distinguish species/group [48]. The PTP method was used through the online tool (http://species.h-its.org/, accessed on 1 October 2013) with default settings, and the output of the ML and BI optimization algorithms was reported. An ultrametric tree was constructed based on Bayesian analysis using the penalized likelihood (PL) method and the truncated Newton (TN) algorithm on r8s software v 1.71 [49]. Then, a GMYC model with multiple thresholds was run on the ultrametric gene tree with R software v 2.15 [50,51] to identify potential COI/ITS1 group representing independently evolving entities. In the case of discordance in the amount of splitting, we chose to keep the smallest number of entities to avoid over-splitting the species complex [46].

Because different COI/ITS1 groups in the *B. calyciflorus* species complex are difficult to visually discriminate under a microscope [52], the density of each COI/ITS1 group in the *B. calyciflorus* species complex at each sampling date in each lake was calculated as *d_i_* = *d_c_* × *p_i_*, where *d_i_*, *d_c_*, and *p_i_* indicates the density of the *i*th COI/ITS1 group, the density of the species complex, and the relative frequency of *i*th COI/ITS1 group, respectively. The relative frequency of each COI/ITS1 group (*p_i_*) was derived from the DNA data analysis and was calculated with *p_i_* = *n_i_*/*n*, where *n_i_* and *n* represents the individual numbers of the *i*th COI/ITS1 group and the individual numbers of the species complex, respectively [8,16,31].

### 2.5. Data Analysis

In order to investigate the most influential variables among environmental variables (temperature, pH, DO, chl-*a* concentration, and the densities of *Asplanchna*, copepods, and cladocerans) in each of the three lakes, a principal component analysis (PCA) was carried out based on the covariance matrix of these variables using the program PAST [32]. Three variables (the densities of *Asplanchna*, copepods, and cladocerans) in both Lake Yunlong and Lake Jinghu, and five variables (water temperature, TP and dissolved oxygen concentrations, and the densities of *Asplanchna* and cladocerans) in Lake Jinniu were very strongly skewed and were transformed to lg (*x* + 1) or lg *x* (only for water temperature) [33]. After the PCA analysis, the frequency pie chart of the COI/ITS1 group in the *B. calyciflorus* species complex for each sampling was placed at the respective sampling position on the two-dimensional space. Thus, the relationship was determined between the COI/ITS1 group frequency and each environmental variable [8,31]. Subsequently, the effects of the most influential variables on the relative frequency and density of each COI/ITS1 group were measured using a generalized linear model (GLM) analysis of deviance with a Poisson distribution and a logit link function in R2.13.0 [53].

## 3. Results

### 3.1. Temporal Variation in Environmental Variables

Throughout the sampling period in Lake Yunlong, Lake Jinghu, and Lake Jinniu, the highest water temperatures occurred in June 2019, July 2019, and October 2018, and the lowest water temperatures occurred in February 2019, November and December 2018, and December 2018, respectively (Table 1).

Among the three lakes, the amplitude in fluctuation of pH was the greatest in Lake Yunlong and the smallest in Lake Jiuniu; the opposite was true for TP and NH_4_^+^–N concentrations and densities of cladocerans and copepods. Amplitudes in the fluctuation of chl-a concentration and TN content were the greatest in Lake Jiuniu and the smallest in Lake Jinghu; the opposite was true for the density of the rotifer *Asplanchna*. Amplitude in the fluctuation of dissolved oxygen concentration was the greatest in Lake Yunlong and the smallest in Lake Jinghu (Table 1, Figure 1).

### 3.2. Sequence Variation, Phylogenetic Relationships, and COI/ITS1 Group Diagnosis

A 632-bp fragment of mtCOI and a 234-bp fragment of nuITS1 were generated from 790 individuals within the *B. calyciflorus* species complex collected from the three lakes. All mtCOI and nuITS1 sequences have been deposited in GenBank (accession numbers ON114186-ON114975 and ON119425-ON120215, respectively). In 790 mtCOI sequences, a total of 316 polymorphic sites, including 217 parsimony informative sites, defined 44 shared haplotypes. In 790 nuITS1 sequences, a total of 62 polymorphic sites, with 46 parsimony informative sites, resulted in 22 shared haplotypes. Most haplotypes occurred in single samples in each lake at a given time, but a few haplotypes were shared by two or more samples from two or three lakes (Appendix A).

The maximum-likelihood phylogenetic tree reconstructed using the COI sequences showed that the *B. calyciflorus* species complex consisted of five distinct groups (“6”, “11”, and “13–15”; Figure 2), and those reconstructed using the ITS1 sequences showed that the *B. calyciflorus* species complex consisted of three distinct groups (“A”, “C”, and “D”; i.e., three species: *B. dorcas*, *B. calyciflorus* s.s., and *B. fernandoi*, respectively; Figure 2). Based on the mtCOI dataset, the GMYC model gave optimal solutions for 11 evolving entities. The estimate of five groups was provided using the ABGD model with 0.022 prior maximal distance, and the estimated number of eleven groups was provided using the PTP method. Based on the nuITS1 dataset, the GMYC model gave optimal solutions for eight evolving entities. The estimate of three groups was provided using the ABGD model with 0.022 prior maximal distance, and the estimate of three groups was provided using the PTP method. According to Papakostas et al. [44], we chose the more conservative number of entities. Hence, the five COI clades were identified as five COI groups and named “6”, “11”, “13”, “14”, and “15”, following Papakostas et al. [44]. The three ITS1 clades were also identified as three distinct groups (“A”, “C”, and “D”; i.e., three species: *B. dorcas*, *B. calyciflorus* s.s., and *B. fernandoi*, respectively). The COI groups “6”, “11”, “13”, “14”, and “15” comprised fourteen, forty-three, three, one, and thirty-four COI haplotypes, and ITS1 groups “A” (*B. dorcas*), “C” (*B. calyciflorus* s.s.), and “D” (*B. fernandoi*) comprised three, forty-five, and nine ITS1 haplotypes, respectively (Figure 2).

### 3.3. Temporal Distributions, Relative Frequencies, and Densities of COI/ITS1 Groups

In Yunlong, Jinghu, and Jinniu lakes, the *B. calyciflorus* species complex comprised five COI groups “6”, “11”, “13”, “14”, and “15”, or three ITS1 groups “A” (*B. dorcas*), “C” (*B. calyciflorus* s.s.), and “D” (*B. fernandoi*) (Figure 3).

In Lake Yunlong, the *B. calyciflorus* species complex comprised four COI groups “11”, “13”, “14”, and “15”, or two ITS1 groups “C” (*B. calyciflorus* s.s.) and “D” (*B. fernandoi*). COI groups “11” and “15”, or ITS1 groups “C” and “D”, underwent clear seasonal successions. In October 2018, the *B. calyciflorus* species complex comprised exclusively COI group “11” or ITS1 group “C”. In November 2018, COI groups “15” or ITS1 group “D” appeared and overlapped with “11” or “C”. From December 2018 to April 2019, COI group “15” displaced “11” alone or with “14” and/or “13”, and ITS1 group “D” replaced “C”. During the overlapping periods of COI group “15” and other COI groups, or ITS1 groups “D” and “C”, the relative frequency and density of COI group “15”, or ITS1 group “D” was always higher. From May 2019 on, COI group “15”, or ITS1 group “D” was displaced by “11” or “C” (Figure 3 and Figure 4).

In Lake Jinghu, the species complex comprised four COI groups “6”, “11”, “14”, and “15”, or three ITS1 groups “A” (*B. dorcas*), “C” (*B. calyciflorus* s.s.) and “D” (*B. fernandoi*). COI groups “15”, “11”, and “6”, or ITS1 groups “D”, “C”, and “A” underwent clear seasonal successions. In October 2018, the species complex comprised exclusively COI group “11” or ITS1 group “C”. In November 2018, COI groups “15”, “14”, and “6”, or ITS1 groups “D” and “A” appeared and overlapped with “11” or “C”. Between December 2018 and March 2019, COI group “15” displaced “11” alone or with “14”, with “15” having a much higher relative frequency and density than “14”; ITS1 group “D” displaced “C”. Between April and May 2019, COI group “11” or ITS1 group “C” displaced “15” or “D” alone or with “6” or “A”, with “11” or “C” having a much higher relative frequency and density than “6” or “A”. Between June and August 2019, COI group “6” or ITS1 group “A” overlapped with “11” or “C” and finally displaced “11” or “C”. During the overlapping period of COI groups “6” and “11”, or ITS1 groups “A” and “C”, “11” or “C” had a lower relative frequency and density than “6” or “A” in June, and the opposite was true in July and September 2019 (Figure 3 and Figure 4).

In Lake Jinniu, the species complex comprised two COI groups, “6” and “11”, or two ITS1 groups, “A” (*B. dorcas*) and “C” (*B. calyciflorus* s.s.). Throughout the sampling period, COI group “11” or ITS1 group “C” existed in the water body. Between November 2018 and February 2019, and in May 2019, COI group “6” or ITS1 “A” appeared and overlapped with “11” or “C”. COI group “6” had a higher relative frequency and density than “11” between December 2018 and January 2019, and the opposite was true in the other months. Between December 2018 and February 2019, ITS1 group “A” had a higher relative frequency and density than “C”, and the opposite was true in November 2018 and May 2019 (Figure 3 and Figure 4).

### 3.4. Effects of Environmental Variables on the Relative Frequencies and Densities of COI/ITS1 Groups

The principal component analysis (PCA) of water environmental variables in Lake Yunlong and Lake Jinghu revealed two factors to explain 99.02% and 99.67% of the total variance, respectively. Water temperature was positively correlated with factor 1 (F_1_, accounting for 94.51% and 83.37% of the data variance in Lake Yunlong and Lake Jinghu, respectively) and factor 2 (F_2_, accounting for 4.51% and 16.30% of the data variance in Lake Yunlong and Lake Jinghu, respectively); chl-a concentration was correlated positively with factor 1 and negatively with factor 2. When the frequency of each group in each sample collected from these two lakes was represented in the space defined by F_1_ and F_2_ scores, respectively, there were straightforward discriminations among cold- and warm-water groups. COI groups “13”, “14”, and “15” and ITS1 group “D” (*B. fernandoi*) could be considered cold-water groups because they were associated with low F_2_ values (low temperature). COI groups “11” and “6” and ITS1 groups “C” (*B. calyciflorus* s.s.) and “A” (*B. dorcas*) could be considered warm-water groups because they were associated with high F_2_ values (high temperature) (Figure 5). The generalized linear model (GLM) analyses showed that in Lake Yunlong, the densities of COI group “11” and ITS1 group “C” were significantly affected by water temperature and chl-a concentration (all *p* < 0.01), and those of COI group “15” and ITS1 group “D” were significantly affected by water temperature, chl-a concentration, and their interaction (all *p* < 0.01). In Lake Jinghu, the densities of COI groups “6”, “11”, and “15”, and ITS1 groups “A”, “C”, and “D” were significantly affected by water temperature, chl-a concentration, and their interaction (all *p* < 0.05) (Table 2).

The PCA of water environmental variables in Lake Jinniu revealed two factors to explain 99.98% of the total variance. Chl-a concentration was correlated positively with factor 1 (F_1_, accounting for 92.18% of the data variance) and negatively with factor 2 (F_2_, accounting for 7.80% of the data variance); copepod density was correlated negatively with factor 1 and positively with factor 2. COI group “6” and ITS1 group “A” (*B. dorcas*) could be considered group/species vulnerable to copepods because they were associated with low F_2_ values (low copepod density), and COI group “11” and ITS1 group “C” (*B. calyciflorus* s.s.) could be considered group/species immune to copepods because they were associated with high F_2_ values (high copepod density) (Figure 5). GLM analyses showed that the densities of COI groups “6” and “11” and ITS1 groups “A” and “C” were significantly affected by copepod density, chl-*a* concentration, and their interaction (all *p* < 0.01) (Table 2).

## 4. Discussion

This study found five mtCOI groups (“15”, “14”, “13”, “11”, and “6”) and three nuITS1groups (“A”, “C”, and “D”; i.e., three species: *B. dorcas*, *B. calyciflorus* s.s., and *B. fernandoi*, respectively) within the *B. calyciflorus* species complex in Yunlong, Jinghu, and Jnniu lakes, which indicated a remarkably mito-nuclear discordance. The cryptic *B. calyciflorus* species (i.e., ITS1 groups) displayed different temporal distribution patterns among the three lakes. In Lake Yunlong, *B. fernandoi* and *B. calyciflorus* s.s. underwent a clear seasonal succession, which was largely attributed to their differential adaptation to water temperature. In Lake Jinghu, *B*. *fernandoi*, *B*. *calyciflorus* s.s., and *B. dorcas* exhibited both seasonal succession and temporal overlap. Seasonal successions were largely attributed to their differential adaptation to temperature, and temporal overlap resulted from their differential responses to algal food concentration. In Lake Jinniu, *B*. *calyciflorus* s.s. persisted throughout the year and overlapped with *B. dorcas* for five months. Temporal overlap resulted from their differential responses to copepod predation.

Mito-nuclear discordance (i.e., discordance between mtDNA and nuclear phylogenies) across taxa is increasingly recognized as a major challenge to species delimitation based on DNA sequence data [54]. With respect to the *B. calyciflorus* complex, mito-nuclear discordances were observed remarkably between mitochondrial and nuclear groups, and species delimitation based on the ITS1 marker has proved to be more reliable predictors of morphological variation than delimitation using the mitochondrial COI gene [31,44,55]. In this study, we found five mtCOI groups and three nuITS1 groups within the *B. calyciflorus* species complex in the three lakes that had been sequenced for both the COI and ITS1 markers, which indicated a remarkably mito-nuclear discordance. Mito-nuclear discordance is often attributed to differences in levels of male and female ongoing gene flow [56] and suggests interspecific gene introgression and hybridization among lineages [57]. Hybridization amongst the species of the *B. calyciflorus* species complex has already been demonstrated [44] and further supported with crossing experiments [58]. Sympatric distribution of species promotes gene introgression/hybridization [59].

Michaloudi et al. reviewed the geographical distribution of the *B. calyciflorus* species complex: *B. calyciflorus* s.s. has a cosmopolitan distribution, whereas *B. dorcas* occurs in Palearctic, Tropical, Oriental, and Australian regions, and *B. elevatus* and *B. fernandoi* are distributed in Palearctic and Oriental regions [26]. Yang et al. found that *B. calyciflorus* s.s. occurs in the Eastern Plain and the Yunnan–Guizhou Plateau in China; *B. dorcas* is restricted to the Eastern Plain; *B. elevatus* occurs in the Eastern Plain, Northeast Plain, Inner Mongolia–Xinjiang Plateau, and Qinghai–Tibetan Plateau; and *B. fernandoi* is distributed in the Eastern Plain, Inner Mongolia–Xinjiang Plateau, and Qinghai–Tibetan Plateau [55]. In this study, *B. calyciflorus* s.s. occurs in all three lakes, but the opposite was true for *B. elevatus*. *B. dorcas* was not detected in the samples from Lake Yunlong, and *B. fernandoi* was not detected in those collected from Lake Jinniu. Considering the short life cycle and fast reproductive ability of these rotifer species, a higher frequency of sampling is necessary in future studies.

Zooplankters dwell in temporally variable habitats where large-scale changes in their abiotic and biotic environments may impact population demographics and genetic structure. Consequently, many zooplankton species occur during restricted seasons, and sympatric species can occur in seasonal succession [60]. For example, some cryptic *B. plicatilis* species in ponds and lakes undergo seasonal succession, although others overlap for short or long periods [8,14,15,16,17,18]. *B. fernandoi*, *B. calyciforus* s.s., and *B. dorcas* within the *B. calyciflorus* species complex in Lake Tingtang also display seasonal succession [31]. In this study, *B*. *calyciflorus* s.s. and *B*. *fernandoi* in Lake Yunlong displayed seasonal succession; *B*. *fernandoi*, *B. dorcas*, and *B*. *calyciflorus* s.s. in Lake Jinghu displayed seasonal successions, although *B. dorcas* and *B*. *calyciflorus* s.s. overlap for a long period. *B*. *calyciflorus* s.s. and *B. dorcas* in Lake Jinniu did not exhibit seasonal succession. These results supported the hypothesis that the temporal distribution pattern of the cryptic *B. calyciforus* species varies with different climate zones. It should be noted that following the framework provided by the theory of coexistence in fluctuating environments [61,62], the short-term disappearance of *B. dorcas* and *B. calyciforus* s.s. from the water column of Lake Jinghu (in May and August 2019, respectively) did not necessarily involve species exclusion.

Because of their short generation times and complex life cycles, the seasonal succession of zooplankton species often correlates with abiotic conditions, indicating certain levels of ecological specialization [7,13]. Seasonal succession of some cryptic *B. plicatilis* species in coastal Mediterranean ponds is largely explained by their differential adaptation to combinations of salinity and temperature [8,14,15,16,19,20,21], and such succession in an inland salt lake (Lake Koronia, Greece) is because of differential ecological preferences to water temperature [18]. Seasonal succession of *B. fernandoi*, *B. calyciforus* s.s., and *B. dorcas* in Lake Tingtang is also explained by differences in their adaptation to water temperature [31,32,33]. Identical results were obtained in this study. *B. fernandoi* had a preference for lower water temperatures (3.2–18.9 °C in Lake Yunlong and 5.6–16.5 °C in Lake Jinghu), but the opposite was true for *B. calyciforus* s.s. and *B. dorcas* (16.7–28.7 °C in Lake Yunlong, 19.4–34.3 °C in Lake Jinghu, and 17.0–29.0 °C in Lake Jinniu). We, therefore, considered *B. fernandoi* as a cold-water species and *B. calyciforus* s.s. and *B. dorcas* as warm-water species, corresponding to heat-sensitive and heat-tolerant species, respectively [33].

How competing species coexist is a fundamental ecological question [21]. Two hypotheses have been advanced to explain the temporal overlap of the cryptic *B. plicatilis* and *B. calyciflorus* species: (i) that sufficient resources and the natural environmental fluctuations allow these species to coexist [8,30]; (ii) that the stable coexistence of potentially strongly competitive cryptic species may be a result of their differential responses to environmental conditions such as salinity [8,14,15,16,17,18,19,20,21] and oxygen availability [22], resource partitioning and differential vulnerability to predators [23,24,25,27,30,31]. This study showed that the synchronous coexistence of *B*. *calyciflorus* s. s. and *B. dorcas* in Lake Jinghu results from their differential responses to algal food concentration and also because of differential responses to copepod predation in Lake Jinniu. These results supported the hypothesis that the mechanisms underlying the temporal overlap of potentially strong competitors are different between climate zones.

## 5. Conclusions

In Lake Yunlong, *B*. *fernandoi* and *B*. *calyciflorus* s.s. underwent a clear seasonal succession, which was largely attributed to their differential adaptation to water temperature. In Lake Jinghu, *B*. *fernandoi*, *B*. *calyciflorus* s.s., and *B. dorcas* exhibited both seasonal succession and temporal overlap. Seasonal successions were largely attributed to their differential adaptation to temperature, and temporal overlap resulted from their differential responses to algal food concentration. In Lake Jinniu, *B*. *calyciflorus* s.s. persisted throughout the year and overlapped with *B. dorcas* for five months. Temporal overlap resulted from their differential responses to copepod predation. These results indicated that the temporal distribution pattern of the cryptic *B. calyciforus* species and the mechanism that allows competitor coexistence vary with different climate zones. Further studies of additional lakes in each climatic zone are essential to know the generality of this conclusion.

## Figures and Tables

**Figure 1 animals-14-00244-f001:**
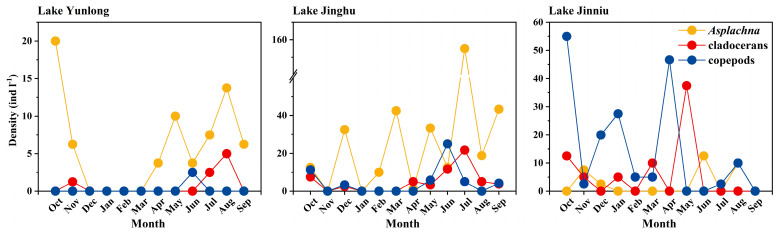
Temporal fluctuations of the densities of *Asplanchna*, cladocerans, and copepods in Lake Yunlong, Lake Jinghu, and Lake Jinniu.

**Figure 2 animals-14-00244-f002:**
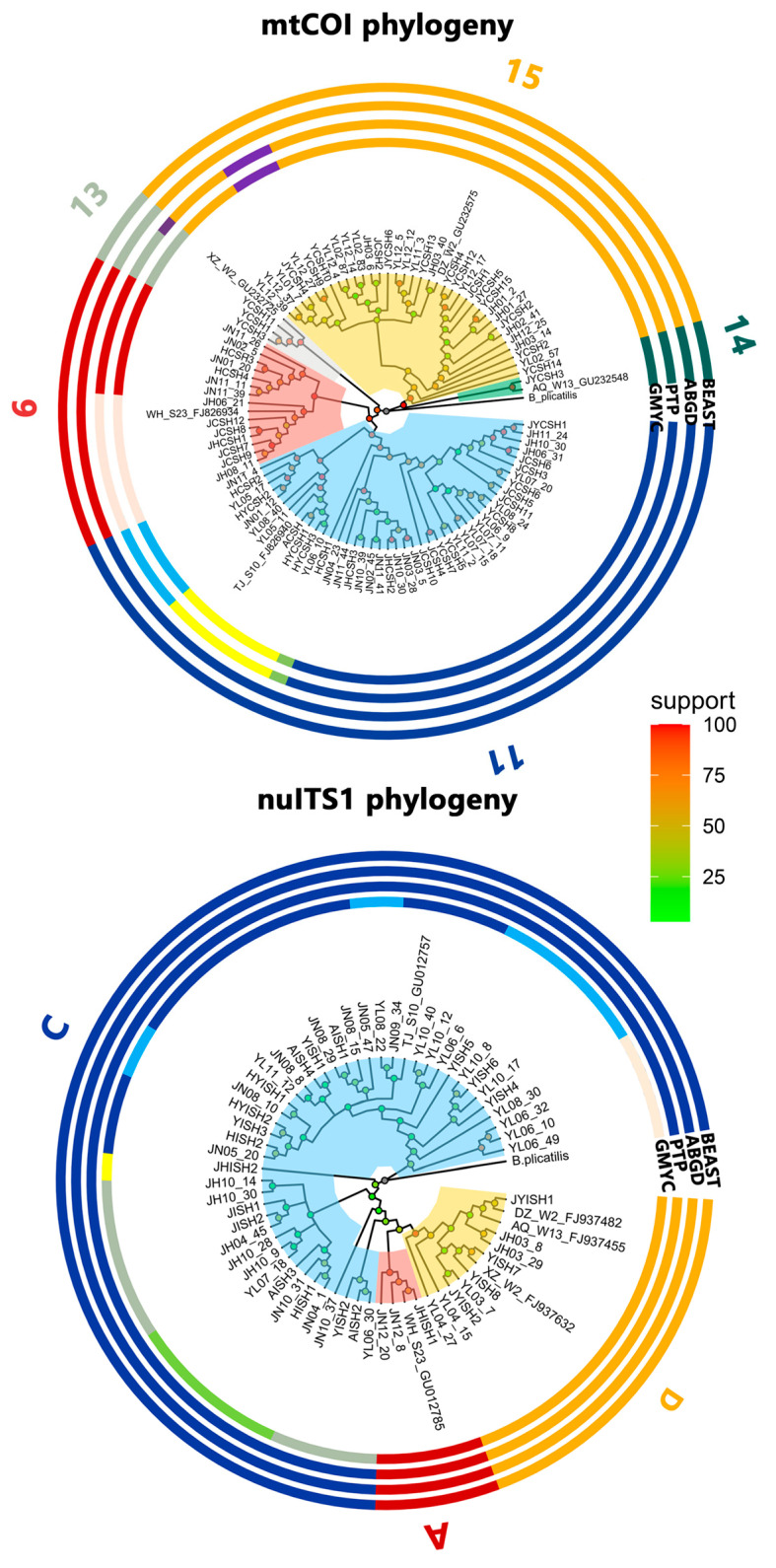
The maximum-likelihood phylogenetic trees and DNA taxonomy results of the Brachionus calyciflorus species complex based on the mtCOI and nuITS1 sequences from Lake Yunlong, Lake Jinghu, and Lake Jinniu. “A”, “C”, and “D” represent *B. dorcas*, *B. calyciflorus* s.s., and *B. fernandoi*, respectively.

**Figure 3 animals-14-00244-f003:**
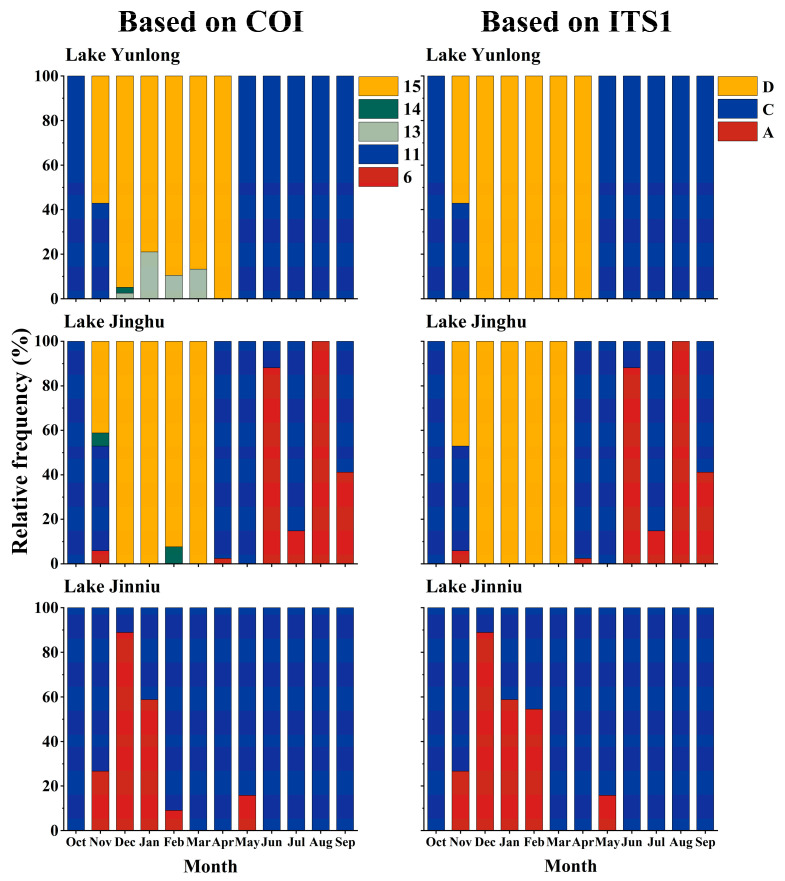
Relative frequencies of cryptic Brachionus calyciflorus groups in Lake Yunlong, Lake Jinghu, and Lake Jinniu. “A”, “C”, and “D” represent *B. dorcas*, *B. calyciflorus* s.s., and *B. fernandoi*, respectively.

**Figure 4 animals-14-00244-f004:**
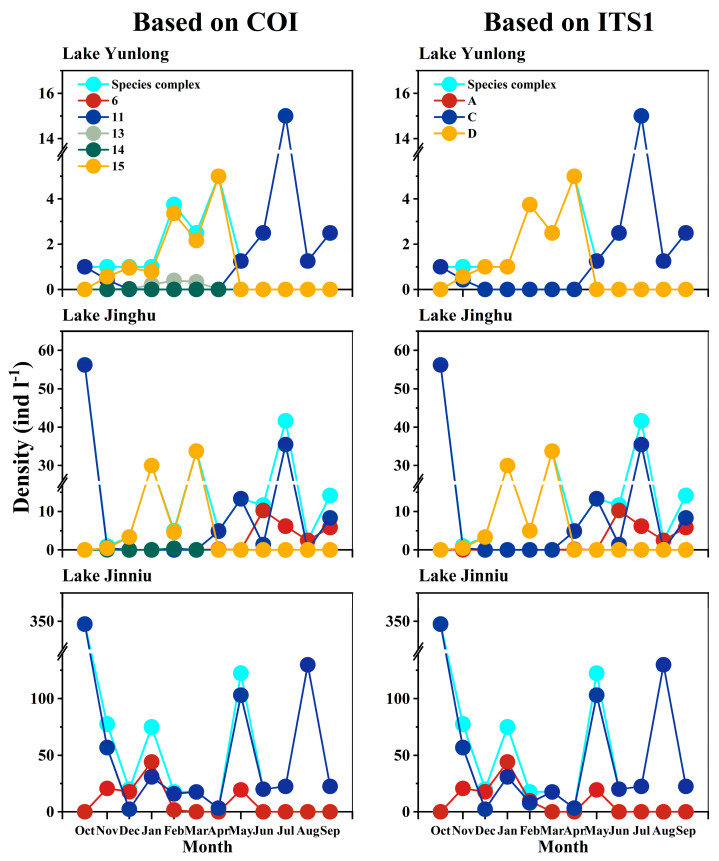
Densities of cryptic Brachionus calyciflorus groups and the *B. calyciflorus* species complex in Lake Yunlong, Lake Jinghu, and Lake Jinniu. “A”, “C”, and “D” represent *B. dorcas*, *B. calyciflorus* s.s., and *B. fernandoi*, respectively.

**Figure 5 animals-14-00244-f005:**
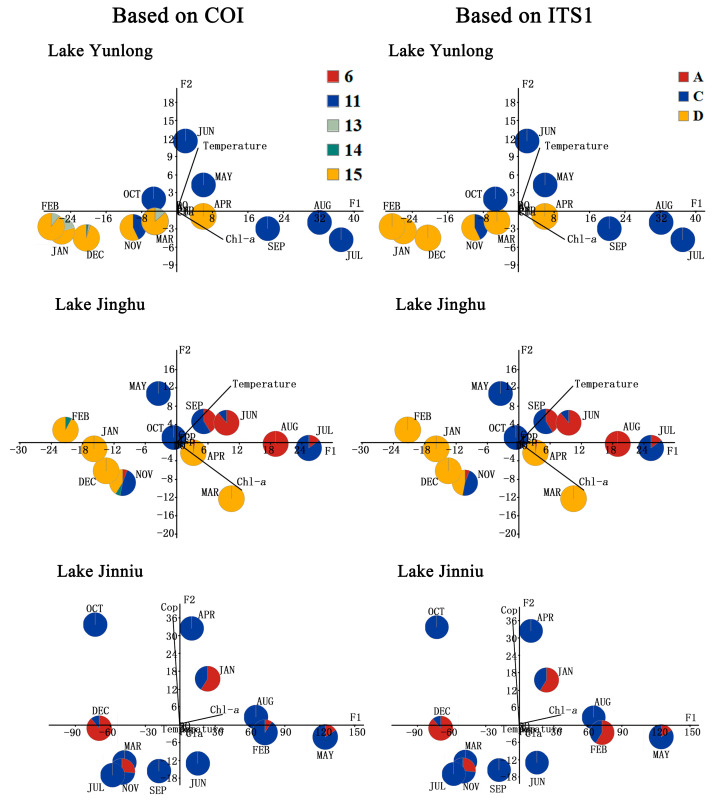
Principal component analyses on the environmental variables (temperature, pH, DO, chl-*a* concentration, and the densities of *Asplanchna*, copepods, and cladocerans) in Lake Yunlong, Lake Jinghu, and Lake Jinniu. Three variables (the densities of *Asplanchna*, copepods, and cladocerans) in both Lake Yunlong and Lake Jinghu, and five variables (water temperature, TP and dissolved oxygen concentrations, and the densities of *Asplanchna* and cladocerans) in Lake Jinniu were very strongly skewed and were transformed to lg (*x* + 1) or lg *x* (only for water temperature). “DO” represents dissolved oxygen, “Asp.” represents *Asplanchna*, “Cop.” represents copepods, and “Cla.” represents cladocerans. “A”, “C”, and “D” represent *B. dorcas*, *B. calyciflorus* s.s., and *B. fernandoi*, respectively.

**Table 1 animals-14-00244-t001:** Summary information for each sample.

Sample Code	Collection Date	Number of Sequenced Individuals	Tem (°C)	pH	Chl-*a* (μg L^−1^)	DO (mg L^−1^)	TN (mg L^−1^)	TP (mg L^−1^)	NH4^+^–N (mg L^−1^)
Lake Yunlong									
YL10	Oct 2018	23	16.7	7.73	25.12	8.90	1.29	0.133	0.31
YL11	Nov 2018	7	10.9	8.16	22.39	1.42	1.61	0.090	0.33
YL12	Dec 2018	40	4.9	8.2	13.65	3.90	1.48	0.067	0.32
YL01	Jan 2019	19	3.7	8.85	8.19	2.14	2.32	0.046	0.41
YL02	Feb 2019	19	3.2	8.77	5.46	2.30	1.96	0.057	0.34
YL03	Mar 2019	15	14.0	8.6	26.75	1.97	1.96	0.058	0.46
YL04	Apr 2019	16	18.9	6.94	36.04	1.35	2.27	0.062	0.21
YL05	May 2019	30	23.6	8.92	33.85	1.74	1.91	0.065	0.33
YL06	Jun 2019	33	28.7	9.06	27.30	1.94	1.99	0.073	0.36
YL07	Jul 2019	26	28.3	8.92	66.07	1.13	1.37	0.092	0.40
YL08	Aug 2019	42	28.6	9.22	60.61	1.52	2.02	0.069	0.41
YL09	Sep 2019	20	23.0	8.48	50.23	1.53	1.90	0.095	0.44
Lake Jinghu									
JH10	Oct 2018	18	19.5	8.72	20.75	1.38	0.96	0.087	0.04
JH11	Nov 2018	17	5.6	8.93	19.66	2.66	0.93	0.073	0.04
JH12	Dec 2018	16	5.6	7.90	15.29	2.30	1.11	0.048	0.04
JH01	Jan 2019	20	8.0	9.16	10.37	2.35	0.92	0.049	0.05
JH02	Feb 2019	13	7.6	8.41	3.82	1.80	0.67	0.030	0.06
JH03	Mar 2019	20	16.5	8.85	37.67	1.21	1.16	0.086	0.05
JH04	Apr 2019	41	19.4	8.59	25.66	1.97	0.70	0.058	0.06
JH05	May 2019	15	25.2	9.19	12.01	1.44	0.84	0.082	0.15
JH06	Jun 2019	43	28.5	9.51	26.21	1.75	1.35	0.087	0.33
JH07	Jul 2019	27	34.3	8.93	42.04	0.77	0.68	0.031	0.10
JH08	Aug 2019	18	31.0	8.61	36.58	1.06	1.50	0.125	0.14
JH09	Sep 2019	17	25.9	8.49	22.93	1.04	1.06	0.078	0.31
Lake Jinniu									
JN10	Oct 2018	9	29.0	6.56	24.70	1.83	3.37	0.120	2.38
JN11	Nov 2018	15	26.0	6.60	40.18	0.51	8.73	1.000	1.67
JN12	Dec 2018	18	17.0	6.70	21.83	0.43	12.38	0.920	7.05
JN01	Jan 2019	17	20.0	6.20	115.40	8.06	11.61	0.300	5.65
JN02	Feb 2019	22	23.0	6.70	165.00	1.64	9.86	0.490	1.04
JN03	Mar 2019	38	25.0	6.70	45.80	4.47	4.96	0.180	1.77
JN04	Apr 2019	15	26.0	6.50	105.80	5.80	2.88	0.020	0.34
JN05	May 2019	38	26.0	6.50	215.90	5.34	3.46	0.180	0.26
JN06	Jun 2019	12	25.5	7.12	107.80	4.95	4.15	0.260	2.54
JN07	Jul 2019	9	25.5	6.70	36.48	5.07	6.73	0.090	1.51
JN08	Aug 2019	14	25.5	6.40	156.38	4.82	3.41	0.150	1.64
JN09	Sep 2019	28	25.0	6.40	74.63	6.05	4.02	0.190	1.18

Tem: water temperature, Chl-*a*: chlorophyll *a* content, DO: dissolved oxygen concentration.

**Table 2 animals-14-00244-t002:** Effects of principal environmental variables on densities of the main mtCOI/nuITS1 groups in Lake Yunlong, Lake Jinghu, and Lake Jinniu using GLMs.

Groups		Lake Yunlong			Lake Jinghu			Lake Jingniu		
		Tem (A)	Chl-*a* (B)	A × B	Tem (A)	Chl-*a* (B)	A × B	Cop (A)	Chl-*a* (B)	A × B
“6”	*z*	-	-	-	30.01	23.26	−25.54	−13.405	5.603	23.279
	*P*	-	-	-	<2 × 10^−16^ ***	<2 × 10^−16^ ***	<2 × 10^−16^ ***	<2 × 10^−16^ ***	<2.11 × 10^−8^ ***	<2 × 10^−16^ ***
“11”	*z*	8.978	4.923	−0.916	36.892	−1.924	−4.899	273.1	140.3	−171.5
	*P*	<2 × 10^−16^ ***	8.53 × 10^−7^ ***	0.36	<2 × 10^−16^ ***	0.0544	9.65 × 10^−7^ ***	<2 × 10^−16^ ***	<2 × 10^−16^ ***	<2 × 10^−16^ ***
“13”	*z*	−0.109	−0.611	−1.312	-	-	-	-	-	-
	*P*	0.913	0.541	0.19	-	-	-	-	-	-
“15”	*z*	−4.526	12.448	−13.961	85.53	55.86	−74.39	-	-	-
	*P*	6.02 × 10^−6^ ***	<2 × 10^−16^ ***	<2 × 10^−16^ ***	<2 × 10^−16^ ***	<2 × 10^−16^ ***	1.06 × 10^−9^ ***	-	-	-
“A”	*z*	-	-	-	30.21	23.30	−25.59	−12.39	14.86	21.52
	*P*	-	-	-	<2 × 10^−16^ ***	<2 × 10^−16^ ***	<2 × 10^−16^ ***	<2 × 10^−16^ ***	<2 × 10^−16^ ***	<2 × 10^−16^ ***
“C”	*z*	8.978	4.923	−0.916	36.576	−2.045	−4.667	273.4	139.4	−171.1
	*P*	<2 × 10^−16^ ***	8.53 × 10^−7^ ***	0.36	<2 × 10^−16^ ***	0.0408 *	3.05 × 10^−6^ ***	<2 × 10^−16^ ***	<2 × 10^−16^ ***	<2 × 10^−16^ ***
“D”	*z*	−4.98	11.81	−13.41	74.21	51.74	−66.63	-	-	-
	*P*	6.35 × 10^−7^ ***	<2 × 10^−16^ ***	<2 × 10^−16^ ***	<2 × 10^−16^ ***	<2 × 10^−16^ ***	<2 × 10^−16^ ***	-	-	-

Tem: water temperature, Chl-*a*: chlorophyll a content, Cop: Copepod density. A: *B. dorcas*, C: *B*. *calyciflorus* s.s., D: *B*. *fernandoi*. *** 0.001, * 0.05.

## Data Availability

The data presented in this study are available in article or Appendix A.

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
