# Peer review of "Temporal Distribution Patterns of Cryptic Brachionus calyciflorus (Rotifera) Species in Relation to Biogeographical Gradient Associated with Latitude"

_animals, 2024, doi:10.3390/ani14020244_

Round 1
Reviewer 1 Report
Comments and Suggestions for Authors
The authors collected zooplankton samples monthly for one year and applied the methods of phylogenetic analysis, DNA taxonomy and principal component analysis to investigate the temporal distribution patterns of cryptic B. calyciflorus species in three lakes in different climate zones in China, and to explore the putative mechanisms for their seasonal succession and/or temporal overlap. Generally speaking, the experimental design is sound, the data is sufficient, the writing is good, and the conclusions is robust. I recommend to accept it after a minor revision. Below are some specific comments:
1. Line 66: There is a letter “n” at the end of this line. Is it “on”?
2. Line 66-74: The cryptic species of B. calyciflorus species complex should be briefly introduced, such as the number of cryptic species found. The sentences from line 68 to line 74 are suggested for discussion.
3. Line 107: “two zooplankton samples” to “two qualitative zooplankton samples”.
4. Line 116-118: The chlorophyll a determination method is not well described.
5. Line 200: Table 1, What does “sample size” mean? Is it the number of individual rotifer B. calyciflorus to be DNA taxonomy?
6. Line 275: What are the common characteristics of B. calyciflorus species complex in the three lakes? Add several sentences to describe the common distribution of the species complex.
7. Line 318: In this section, the lake Jinniu in Fig.5 is not described.
8. Line 378. The number of the table is wrong. It is table 2.
Author Response
Dear reviewers,
Thank you very much for your comments and constructive suggestions on our manuscript, which have greatly improved our manuscript. Following your comments and constructive suggestions, I carefully revised the manuscript again. The revisions were presented as following.
Best regards.
Sincerely yours,
Yi-Long Xi
1. Line 66: There is a letter “n” at the end of this line. Is it “on”?
Re: We changed the letter “n” into “on”.
2. Line 66-74: The cryptic species of B. calyciflorus species complex should be briefly introduced, such as the number of cryptic species found. The sentences from line 68 to line 74 are suggested for discussion.
Re: We added a brief introduction by “that was recently suggested to consist of four species: B. dorcas, B. elevatus, B. calyciflorus s.s. and B. fernandoi [26]” in the first sentence of the third paragraph in Introduction (Page 2).
We removed the sentence “Seasonal succession of B. fernandoi, and B. calyciforus s.s. and B. dorcas is largely attributed to their differential adaptation to temperature [30-32] and temporal overlap of B. calyciforus s.s. and B. dorcas results from their differential responses to algal food concentration”.
3. Line 107: “two zooplankton samples” to “two qualitative zooplankton samples”.
Re: We changed “two zooplankton samples” into “two qualitative zooplankton samples”.
4. Line 116-118: The chlorophyll a determination method is not well described.
Re: We added it by “Integrated lake water was collected using a 5-l modified Van-Dorn sampler (5 L of water from the surface to the bottom at 0.5 m intervals) and then filtered through a 25-μm mesh net. Filtrate was further filtered through Whatman GF/C glass-fiber filters (0.45 µm pore size) to obtain relatively small phytoplankton that might be a food resource for rotifers. The biomass of relatively small phytoplankton was represented by chlorophyll a (Chl-a) concentration. Chl-a content was spectrophotometrically measured after extracting the filters overnight at dark using 90% acetone and calculated without correcting for phaeopigments [35]”. (Page 3)
5. Line 200: Table 1, What does “sample size” mean? Is it the number of individual rotifer B. calyciflorus to be DNA taxonomy?
Re: We changed “sample size” into “number of sequenced individuals”.
6. Line 275: What are the common characteristics of B. calyciflorus species complex in the three lakes? Add several sentences to describe the common distribution of the species complex.
Re: We added a paragraph in Page 8: In Yunlong, Jinghu and Jinniu lakes, the B. calyciflorus species complex comprised five COI groups “6”, “11”, “13”, “14” and “15”, or three ITS1 groups “A” (B. dorcas), “C” (B. calyciflorus s.s.) and “D” (B. fernandoi) (Fig. 3).
7. Line 318: In this section, the lake Jinniu in Fig.5 is not described.
Re: We added a paragraph in Page 10-11: PCA of water environmental variables in Lake Jinniu revealed two factors to explain 99.98% of the total variance. Chl-a concentration was correlated positively with factor 1 (F1, accounting for 92.18% of the data variance), and negatively with factor 2 (F2, accounting for 7.80% of the data variance); copepod density was correlated negatively with factor 1, and positively with factor 2. COI group “6” and ITS1 group “A” (B. dorcas) could be considered group/species vulnerable to copepods because they were associated with low F2 values (low copepod density), and COI group “11” and ITS1 group “C” (B. calyciflorus s.s.) could be considered group/species immune to copepods because they were associated with high F2 values (high copepod density) (Fig. 5). GLM analyses showed that the densities of COI groups “6” and “11”, and ITS1 groups “A” and “C” were significantly affected by copepod density, chl-a concentration and their interaction (all p < 0.01) (Table 2).
8. Line 378. The number of the table is wrong. It is table 2.
Re: We corrected it.
Reviewer 2 Report
Comments and Suggestions for Authors
This is one of those rather rare manuscripts that is so good that it leaves nothing to criticize. It again highlights that much more research is needed to get a better understanding of cryptic rotifer diversity and seasonal successio n driven by responses to environmental parameters, and allowing for competitor coexistence varying with different climatic zones.
There are very few little typos or writing glitches:
- l. 16: In tropical... > However, in tropical...
- l. 66: n > on
- l. 26: Asplanchna : italics
- l. 333: delete space
- Fig. 5: add A, C, D in legend
- l. 431: replace Gabaldon... by [20-21]
- References: check year of publication in bold (e.g. l. 490, 493, 495, 521, 531, 550 also no italics),
-
Author Response
Dear reviewers,
Thank you very much for your comments and constructive suggestions on our manuscript, which have greatly improved our manuscript. Following your comments and constructive suggestions, I carefully revised the manuscript again. The revisions were presented as following.
Best regards.
Sincerely yours,
Yi-Long Xi
- l. 16: In tropical... > However, in tropical...
Re: We changed “In tropical lakes, however, …..” into “However, in tropical lakes, …..”.
- l. 66: n > on
Re: We corrected it.
- l. 26: Asplanchna : italics
Re: We corrected it.
- l. 333: delete space
Re: We did it.
- Fig. 5: add A, C, D in legend
Re: We added two sentences in the legend: “A”, “C”, and “D” represent B. dorcas, B. calyciflorus s.s., and B. fernandoi, respectively.
- l. 431: replace Gabaldon... by [20-21]
Re: We replaced it.
- References: check year of publication in bold (e.g. l. 490, 493, 495, 521, 531, 550 also no italics),
Re: We corrected it.
Reviewer 3 Report
Comments and Suggestions for Authors
The authors investigated the seasonal succession and temporal overlap of cryptic Brachionus calyciflorus species in three lakes located in different climatic zones. The use of molecular genetic methods allowed the authors to accurately identify the cryptic species. All seasons of the year were continuously covered, as zooplankton sampling was carried out monthly for the whole year. A thorough analysis of the relationship between the abundance of cryptic species and environmental factors, the availability of food resources, and the pressure of predators was carried out. As a result, the conclusion was drawn that the temporal distribution pattern of cryptic B. calyciforus species and the mechanism allowing competitor coexistence vary with different climate zones. The study is of interest because it demonstrates the possibility and conditions for the coexistence of cryptic species.
The manuscript is well presented, but some points require clarification.
In the legend to Figure 5, the designations DO, pH, Asp, Cla are indicated, but they are not visible on the graphs themselves.
What was the minimum abundance of species during the entire observation period? Looking at Figure 4, the number of Brachionus was very small on some sampling dates (in Jinghu Lake in November, in Jinniu Lake in April). However, three species were identified in Jinghu Lake in November based on ITS1.
The manuscript can be accepted after a little revision.
Author Response
Dear reviewers,
Thank you very much for your comments and constructive suggestions on our manuscript, which have greatly improved our manuscript. Following your comments and constructive suggestions, I carefully revised the manuscript again. The revisions were presented as following.
Best regards.
Sincerely yours,
Yi-Long Xi
In the legend to Figure 5, the designations DO, pH, Asp, Cla are indicated, but they are not visible on the graphs themselves.
Re: Because they accounted for very low percentages of the data variance in three lakes.
What was the minimum abundance of species during the entire observation period? Looking at Figure 4, the number of Brachionus was very small on some sampling dates (in Jinghu Lake in November, in Jinniu Lake in April). However, three species were identified in Jinghu Lake in November based on ITS1.
Re: The minimum abundance of species during the entire observation period was close to 0.01 ind l-1. This density estimate was calculated by direct counts of females from two quantitative zooplankton samples that were obtained from two fixed sites of each lake by filtering (25 μm mesh net) two samples of integrated lake water (5 L of water from the surface to the bottom at 0.5 m intervals). However, mtCOI and nuITS1 groups were identified by the sequences of individuals that were collected at the fixed sites of each lake in several hauls using a 25-µm plankton net, and thus more individuals were obtained.